# Inhibiting the Priming for Cancer in Li-Fraumeni Syndrome

**DOI:** 10.3390/cancers14071621

**Published:** 2022-03-23

**Authors:** Pan Pantziarka, Sarah Blagden

**Affiliations:** 1The George Pantziarka TP53 Trust, London KT1 2JP, UK; 2The Anti-Cancer Fund, Brusselsesteenweg 11, 1860 Meise, Belgium; 3Department of Oncology, University of Oxford, Oxford OX3 7DQ, UK; sarah.blagden@oncology.ox.ac.uk

**Keywords:** Li-Fraumeni Syndrome, TP53, drug re-purposing, pre-cancer niche, cancer pre-disposition

## Abstract

**Simple Summary:**

Li-Fraumeni Syndrome (LFS) is a rare cancer pre-disposition syndrome associated with a germline mutation in the TP53 tumour suppressor gene. People with LFS have a 90% chance of suffering one or more cancers in their lifetime. No treatments exist to reduce this cancer risk. This paper reviews the evidence for how cancers start in people with LFS and proposes that a series of commonly used non-cancer drugs, including metformin and aspirin, can help reduce that lifetime risk of cancer.

**Abstract:**

The concept of the pre-cancerous niche applies the ‘seed and soil’ theory of metastasis to the initial process of carcinogenesis. TP53 is at the nexus of this process and, in the context of Li-Fraumeni Syndrome (LFS), is a key determinant of the conditions in which cancers are formed and progress. Important factors in the creation of the pre-cancerous niche include disrupted tissue homeostasis, cellular metabolism and chronic inflammation. While druggability of TP53 remains a challenge, there is evidence that drug re-purposing may be able to address aspects of pre-cancerous niche formation and thereby reduce the risk of cancer in individuals with LFS.

## 1. Introduction

Li-Fraumeni Syndrome (LFS) is a rare autosomal dominant genetic condition that pre-disposes sufferers to develop one or more cancers [1,2]. It is associated with pathogenic germ-line variants in TP53 and has an estimated penetrance of 80–90%, higher than in other cancer pre-disposition syndromes, for example that associated with BRCA1 or BRCA2 [2,3,4]. LFS is associated with a range of cancers including bone and soft tissue sarcomas, early onset breast cancer, choroid plexus carcinoma and adrenocortical carcinoma [1,4]. Although these are considered the ‘signature’ cancers associated with a germline TP53 mutation, it should be noted that LFS pre-disposes one to a far wider range of cancers, including leukaemia, lung cancer and other more common malignancies. In addition to an unusual range of cancer types, there also appears to be a temporal component to cancer incidence, with peaks in childhood and again in the early-30s, the latter associated with the high rate of breast cancer in women [4]. The risk of developing subsequent primary cancers after the first are also very high, with data showing that 49% of people with LFS go on to develop one or more primaries within a median of 10 years [5].

While most people with LFS have inherited a pathogenic TP53 variant, there are also de novo cases, with estimates ranging up to 20% of diagnosed cases [6]. However, such estimates may be subject to detection bias as lower penetrant de novo variants are detected at a lower rate than highly penetrant variants and inherited cases.

A number of different diagnostic criteria exist for LFS and related conditions such as Li-Fraumeni-like Syndrome [6,7,8]. Indeed, recently there has been a proposal to introduce the concept of a Li-Fraumeni Spectrum that includes incidental findings of lower penetrance variants, a broader classification of heritable TP53-related cancer (*hTP53rc*) syndromes and those who have ‘phenotypic LFS’—that is, they have the cancer incidence and family history but without a known genetic driver [8,9].

A number of studies have shown that active surveillance of LFS patients is able to identify pre-symptomatic malignancies and is associated with improved survival [10,11,12]. Prophylactic double mastectomy is also available to women to reduce the elevated risk of early onset breast cancer. Whole-body MRI is at the core of the surveillance protocols which have been adopted across the world in recent years [8,10,13,14]. Notable also is the emergence of a number of active patient advocacy organisations in recent years including the George Pantziarka TP53 Trust in the United Kingdom, Living LFS in the US and the Li-Fraumeni Syndrome Association in the United States and with many chapters across the world.

Lessons learned from cancer prevention and treatment in people with LFS may also generalise more widely. Other cancer pre-disposition syndromes, with different genetic drivers, may share some aspects of the hypothesis explored in this paper, and may therefore benefit from similar prevention approaches. In addition, it remains true that TP53 is the most common somatic mutation in cancer and is particularly prevalent in cancers such as non-small cell lung cancer, in head and neck cancers and in many recurrent/refractory cancers such as testicular cancer. Working with LFS may therefore have wider significance to oncology at large.

While much research activity in LFS remains focused on elucidating the relationship between specific TP53 variants and penetrance, the range and timing of cancer incidence and other epidemiological variables, there is also an increasing interest in developing medical interventions which may act to modulate cancer risk. The success, or otherwise, of drug interventions to reduce cancer incidence in LFS rests in part on developing further our understanding of the process of carcinogenesis in LFS and exploring how aberrant p53 activity interacts with host factors to facilitate the development of malignancy. This paper will explore the hypothesis that cancer incidence in LFS involves multiple aspects of p53 activity over and above explicit tumour suppression via apoptosis. It will also outline the data supporting the clinical exploration of a number of drug candidates for cancer prevention in people with LFS.

## 2. The Pre-Cancer Niche

The concept of the pre-metastatic niche, proposed by David Lyden and colleagues, posits that the process of metastasis depends as much on the preparation of the receptive and supportive micro-environment at the metastatic site as it does on the properties of the metastatic cell itself [15,16]. Crucially, the process depends on factors secreted by the primary tumour to prime the niche sites to create a permissive environment for the growth of circulating tumour cells which can preferentially home to the niche and take root there. An extension of this idea describes the evolution of the primary cancer niche as a multi-step process of carcinogenesis [17].

The pre-cancerous niche hypothesis proposes that a similar process precedes the establishment of primary tumours in LFS [18]. In particular, the hypothesis suggests that pathogenic germline TP53 variants facilitate the creation of these pre-malignant niches. Key drivers for this process are: chronic inflammation and oxidative stress; pro-angiogenic signalling; immune dysregulation; metabolic plasticity; and tissue-specific interactions—with aberrant p53 as a central driver of the process. Furthermore, the pre-cancerous niche itself may cause additional mutational events to occur in local cells which, together with a defective apoptotic apparatus, ensures that cancer initiation takes place in an environment that is permissive and supportive of transformed cells.

Certain phenotypic features that are common to healthy (i.e., non-cancer-carrying), people with LFS add support to this hypothesis. The relationship between chronic inflammation, oxidative stress and p53 signalling is well characterised in numerous pre-cancerous or inflammatory conditions [19,20,21,22]. It is also known that cancer-free LFS sufferers exhibit clinical signs of increased levels of oxidative stress compared to a paired group of non-affected family members (i.e., without TP53 mutations) [23]. Furthermore, mutant p53 has been shown to fine-tune anti-oxidant responses, via NRF2, to support the survival of transformed cells [24].

A pro-angiogenic micro-environment is also a factor in the pre-cancerous niche. Fibroblasts derived from LFS patients confirm that loss of the wild-type p53 allele is sufficient to decrease TSP-1 expression and an increase in VEGF [25]. There is also some evidence that gain-of-function (GOF) TP53 mutations (including R175H and R273H common in people with LFS) have been shown to have tumour angiogenesis-promoting activity [26,27].

Metabolic plasticity is one of the hallmarks of cancer in which p53 signalling plays a central role [28]. Evidence from many cancer types, including from some ‘core’ LFS cancers such as osteosarcoma [29], show the emergence of complex metabolic pockets or compartments within tumours and stroma, including Warburg and reverse Warburg phenotypes, such that metabolites are shuttled between compartments in a process of metabolic adaptation [30,31,32]. In the case of LFS, it has been proposed that such a complex evolutionary process, driven in large part by increased oxidative stress and aberrant p53, is a major factor in carcinogenesis [33]. Notably, a key marker for this multi-compartment metabolic phenotype is loss of stromal cav-1 expression [34], and this finding has been confirmed in people with LFS compared to non-affected family members [35]. Other studies have shown that people with LFS have increased oxidative metabolism compared to non-carriers, a finding in line with data from murine models of LFS [36].

To date, there has been no published analysis examining differences in immune responses between family members harbouring pathogenic TP53 variants and related wild-type carriers. However, the role of p53 in the immune response is complex, with both direct and indirect effects reported [37]. A recent review summarises the data on the impact of mutant p53 on immune dysfunction, showing that it contributes to the creation of a pro-tumour micro-environment, particularly via up-regulation of NF-kB [38]. Mutant p53 also targets toll-like receptor activity in response to chronic inflammation and increased oxidative stress [39,40]. Indirect effects on immunity arise from the metabolic changes induced by mutant p53, for example via increased acidity in the micro-environment.

Given the very specific pattern of LFS cancers, and the fact that the same pathogenic variants within families may manifest in a wide range of cancer types, it is difficult to correlate specific variants with clinical variability [41]. Tissue-specific factors may play a role here. For example, a comparison of breast adipose tissue between women with and without LFS showed a statistically significantly increased aromatase expression in the LFS women [42]. Furthermore, this study also showed that prostaglandin E2 (PGE2), a key inflammatory factor, acts as a negative regulator of p53. This finding may in part explain the distinct pattern of breast cancer phenotypes in women with LFS, with data showing that 84% of invasive tumours were hormone-responsive (ER and/or PR), with a majority of these also being positive for Her2/neu, figures which are higher than for the non-LFS population [43].

Together, these various factors, summarised in Figure 1, combine to form pre-cancerous niches which can both drive malignant transformation in individual cells and provide a supportive environment for these transformed cells to proliferate and ultimately form tumours [18]. Again, evidence from non-cancer-bearing individuals with LFS supports such a hypothesis. A comparison of telomere length of people with LFS compared to non-affected family members shows that TP53 mutation carriers have shorter telomeres [44,45], which may be related to the age of cancer onset [46,47]. We posit that telomere attrition, a process that is exacerbated by the oxidative and other cellular stresses previously described, eventually leads to telomere crisis and consequent DNA damage and, hence, malignant transformation, as shown in Figure 2. Alternatively, the high levels of oxidative stress cause additional DNA damage, including loss of heterozygosity, and initiate malignant transformation. There is supporting evidence that cells from LFS patients display greater levels of DNA damage (chromosomal instability, senescence, etc.) [48,49,50]. One could characterise this process of accelerated telomere attrition and/or oxidative damage due to pathogenic TP53 variants as a form of accelerated host aging—again, this is a known factor in carcinogenesis [51]. DNA methylation age (Horvath age) is an epigenetic measure of aging that correlates with telomere length and can provide an alternative measure of accelerated aging [52]. Recent data show that Horvath age differed from chronological age in people with LFS compared to non-LFS individuals and that the accelerated aging was associated with cancer incidence [53]. It should also be noted that maternal stress is associated with shorter telomere length in children [54]. Given the severe psychological stress associated with LFS and incidences of cancer, it may be that this is a factor in explaining the younger age of cancer onset in succeeding generations in LFS families (genetic anticipation) [47,55].

An interesting experimental illustration of this process comes from an animal model of LFS in which mice differing in Trp53 status were treated with either surgical implantation of a foreign object to induce chronic inflammation or a sham operation [56]. In 30/38 (79%) cases, mice with heterozygous Trp53 developed sarcomas around the implant site at a mean of 46 weeks, compared to one (10%) of the wild-type mice at 56 weeks. No sarcomas developed at the sites of sham operation, and 2/10 (20%) control heterozygous mice (with no implant) also developed sarcomas at a mean age of 80 weeks. Significantly, 90% of implant-induced sarcomas showed loss of heterozygosity, suggesting a causative effect from the chronic inflammation induced by the implant.

One obvious consequence of this view of carcinogenesis in LFS is that altering or ameliorating the pro-tumour elements of the pre-cancer niche may therefore reduce the incidence of cancer.

## 3. Drugging the Undruggable

Lynch syndrome, also known as hereditary non-polyposis colorectal cancer (HNPCC), is another autosomal dominant cancer pre-disposition syndrome that is associated with elevated risks of colorectal, endometrial and other cancers [57,58]. Long-term follow-up of the CAPP2 double-blind randomised placebo-controlled trial of aspirin in people with Lynch syndrome showed that it reduced the incidence of colorectal cancer, with a significantly reduced hazard ratio (HR) of 0.65 (95% CI 0.43–0.97; *p* = 0.035) for aspirin versus placebo, although no effect was shown in non-colorectal cancers [59].

These results are significant not only for patients with Lynch syndrome and clinicians but also for other cancer pre-disposition syndromes. It illustrates the case that the cancer risks arising from genetic pre-disposition can be reduced using medications that impact the pathways associated with carcinogenesis. Furthermore, the efficacy of the drug intervention can be shown through the use of appropriate clinical trial designs.

The specific use of aspirin is further interesting in that it may be paradigmatic of the broader approach to reducing cancer incidence in very high-risk populations. Notably, the approach has been to use an existing, licensed drug rather than to create a new molecular entity—in other words, a development approach based on drug re-purposing. Re-purposing benefits from the use of existing data on drug safety, posology, pharmacokinetics and knowledge of mechanisms of action, in addition to the easy availability of the medications and lower drug costs, particularly in the case of generics [60,61]. In the case of cancer prevention in specific high-risk populations, it is likely that treatments will extend for many years, perhaps even across a patient’s lifetime; therefore, candidate drugs must have long-term safety and tolerability data. Drugs used for chronic diseases or designed for long-term use may therefore have additional benefit as re-purposing candidates. In contrast, there can be no long-term data on newly developed medicines. One consequence is that clinical trials investigating the cancer prevention effects of re-purposed drugs can proceed relatively quickly, as early phase safety trials are not required if the drug is going to be used at a similar dose and schedule to the original use of the drug. It is also worth noting that when treatment extends for many years, drug costs will also become a factor in health technology assessment; therefore, low-cost re-purposing candidates may also be more attractive in this respect.

While the putative benefits of the re-purposing approach are clear, it is still the case that the identification of suitable candidate drugs requires a strong biological rationale. A number of candidate drugs are explored below.

### 3.1. Metformin

Metformin, an anti-diabetic drug that is currently being actively pursued as a drug re-purposing candidate in sporadic cancers, is a major focus of interest in LFS research. It is also one of the most popular of the non-cancer drugs being clinically investigated as a possible cancer treatment. The ReDO_Trials Database (https://www.anti-cancerfund.org/en/redo-trials-db, accessed on 22 November 2021), an online resource that lists active oncology trials investigating the use of non-cancer drugs as anti-cancer agents, lists 131 (16% of the total) active trials that include metformin in an investigation arm (as of 22 November 2021) [62]. Additionally, there are numerous trials of metformin as a cancer-prevention agent in high-risk populations such as people with familial adenomatous polyposis, oral pre-malignant lesions and those suffering from metabolic syndrome at high risk of developing cancer.

Metformin is a highly pleiotropic drug with multiple molecular targets and mechanisms of action which are still being elucidated even for its primary indication of diabetes mellitus (DM) [63]. Its therapeutic effects on DM are due to a reduction in hepatic gluconeogenesis via activation of AMPK and improving insulin sensitivity via AMPK and AMPK-independent mechanisms. In addition to reducing hyperglycaemia and hyperinsulinemia, it also reduces dyslipidaemia via AMPK-mediated inhibition of fatty acid synthesis, inhibits mTOR and reduces circulating insulin-like growth factor (IGF). Metformin also interacts directly with mitochondria via inhibition of mitochondrial respiratory-chain complex 1, thereby reducing reactive oxygen species generation [63,64].

It has also been long established that metformin indirectly targets p53 (via AMPK/LKB1 [65]), with evidence showing that it is selectively toxic to nutrient-deprived p53-deficient cells [66]. Based on the latter finding, the first in vivo explorations of the action of metformin in models of LFS were presented in 2011 [67]. Both p53-null and heterozygous p53^R172H^ models were used, with comparison of overall survival of metformin-treated (5 mg/mL in drinking water) versus untreated mice. Two treatment schedules were used—treatment commenced either early or late in life. In both cases, the data showed that metformin significantly prolonged median overall survival. In the late treatment schedule for heterozygous mice survival, was 20.6 months vs. 15.8 months (*p* = 0.0004), and for p53-null mice, 9.1 months vs. 5.3 months (*p* = 0.0006). In the early-in-life treatment, the results for heterozygous mice were 20.5 months vs. 13 months (*p* = 0.0025); results were not reported for the early-in-life treatment of p53-null mice, as the median survival for metformin-treated mice had not been reached. Biomarker analysis indicated that metformin activated AMPK and inhibited the mTOR pathway in liver tissues.

Evidence of the importance of metabolism and oxidative stress in the actions of p53 also emerged in 2012 with publication of an elegant study by Li et al. that showed that abrogation of the cell-cycle arrest, senescence and apoptotic functions of p53 did not interfere with its tumour suppressive activity [68]. The in vivo study showed that selective acetylation at p53 sites to ablate these functions did not interfere with the antioxidant and metabolic regulation functions and that these were essential in the tumour suppressive activity.

In 2013, Hwang and colleagues compared mitochondrial activity in skeletal muscle cells of people with LFS, non-affected family members and healthy volunteers [36]. In addition to treadmill exercise, the study also analysed tissue samples and corroborated the findings with a murine model of LFS. In all cases, the findings show an increase in mitochondrial function, with increased oxidative phosphorylation of skeletal muscle. Subsequently, the same authors used a mouse model of LFS to show that genetic disruption of mitochondrial respiration, reducing the oxygen consumption rate, increased the cancer-free survival of the mice [69]. Treatment with metformin, at a dose of 1.25 mg/mL in water, increased median and mean cancer-free survival times by 22% and 27%, respectively. Furthermore, the study also assessed changes in mitochondrial function after a short course of metformin in a small number of healthy people with LFS (*n* = 14). Patients were treated for 14 weeks, to a maximum dose of 2000 mg/day, and anti-proliferative and mitochondrial respiration biomarkers were assessed at week 0, 8 and 14 and after six weeks of washout at week 20. The data showed that treatment reduced the oxygen consumption rate, extracellular acidification rate and other markers of mitochondrial function and that after washout these returned to baseline levels. Skeletal muscle phospho-creatine (PCr) recovery kinetics after exercise were also measured, and data showed that metformin treatment increased the PCr recovery time constant, providing in vivo evidence of decreased mitochondrial activity.

A similar schedule of metformin treatment was used in a small (*n* = 26) Phase I study (NCT01981525) to assess the tolerability of daily metformin and the effect on circulating IGF-1, insulin and IGFBP3 in people with LFS [70]. Secondary outcomes included assessment of mitochondrial function in skeletal muscle at baseline and after 8 weeks of metformin. Metformin was found to be tolerable, with a similar profile of adverse events, including grade 1 diarrhoea (50.0%) and nausea (46.2%), as in the general population. Results showed that serum IGF-1 levels were statistically significantly lower when on metformin, as were fasting levels of IGFBP3. Hepatic mitochondrial function was assessed using ^13^C-MBT measurements, and this was decreased while on metformin, suggesting a decrease in hepatic mitochondrial function that returned to baseline levels after washout. The authors of the study conclude that the data provide support for testing the pharmacologic risk-reducing effects metformin in a prospectively designed clinical trial for patients with LFS.

The Metformin in Li-Fraumeni Syndrome (MILI) study is such a prospective clinical trial and has recently been approved for funding by the UK National Institute of Health Research (NIHR) [http://www.tp53.co.uk/2021/11/22/mili-trial-funding-approved/, accessed on 22 November 2021]. This is the first clinical trial to be approved with a cancer incidence reduction end-point in Li-Fraumeni Syndrome. The trial schema is shown in Figure 3.

The trial will recruit 224 LFS adults in the UK with confirmed pathogenic TP53 variants, and patients will be randomised to a metformin plus surveillance arm or to surveillance only. The dose of metformin used will be 2000 mg/day, with a dose escalation period of four weeks. Patients will be followed for five years to assess cumulative cancer-free survival. Secondary outcomes will include comparison of overall survival, the spectrum of cancers and overall quality of life between the two arms of the trial. Similar trials are planned in Canada, the United States and in Germany. As with the MILI trial, each of these trials is being designed so that the data can be meta-analysed to give a global answer to the question of the efficacy, or not, of metformin as a cancer prevention treatment in LFS.

Finally, although not directly relevant to the cancer prevention context, there are two interesting case reports of patients with LFS with cancer who have shown clinically meaningful responses to metformin treatment. In the first case, an infant with LFS and recurrent choroid plexus carcinoma, one of the ‘core’ LFS cancers, was treated with chemotherapy, localised proton beam therapy and subsequently maintenance therapy using a trio of re-purposed non-cancer drugs: metformin, simvastatin and melatonin [71]. The latter treatment, based on morphoproteomic of the recurrent tumour, is well tolerated and is associated with long-term remission. Adrenocortical carcinoma (ACC) is another of the index cancers associated with LFS. An adult patient with LFS and metastatic ACC, a disease with a grim prognosis, was started on metformin following a phenotypic drug screen that identified the related drug phenformin as active against her tumour [72]. Treatment at a relatively low dose of 1000 mg/day was associated with an objective response that lasted for nine months.

Given the strength of the LFS-specific evidence for metformin, we have not reviewed the extensive evidence showing that metformin targets many of the facets of the pre-cancerous niche, for example, positive effects on telomere attrition [73,74], angiogenesis [75,76] and anti-cancer immunity [77]. These additional functions of metformin can be expected to enhance the anti-carcinogenic effects and to reduce cancer risks in people with LFS.

### 3.2. Statins

As with metformin, there is great interest in the use of statin drugs as re-purposing candidates in oncology—primarily as therapeutic options rather than for primary cancer prevention. The ReDO_Trials Database lists 40 active trials (as of 14 December 2021) using statins—with simvastatin and atorvastatin being the candidates with the highest number of trials, although there are also examples of trials investigating the use of fluvastatin, lovastatin, pravastatin and rosuvastatin. Observational data suggest that statin use may be associated with lower cancer incidence in the general population for a number of malignancies, including breast, cervical, lung and colorectal cancers [78,79].

Drugs of the statin class are competitive inhibitors of HMG-CoA reductase, a rate-limiting enzyme in the mevalonate pathway, and thereby act to reduce cholesterol synthesis. There is also increasing evidence that some of the non-canonical actions of these drugs are also important in reducing atherosclerosis, particularly effects on chronic inflammation, endothelial cell function and immunity [80,81,82,83]. In the context of LFS, there is a paucity of relevant human data, and the possible beneficial effects of statins on cancer prevention in this population are based on pre-clinical evidence.

Gain of function p53 mutants such as R175H, which is associated with the core cancers in LFS patients, interact with the mevalonate pathway in a positive feedback loop which confers stabilisation of the mutant p53 proteins, which then up-regulates lipid metabolism [84,85]. A metabolic intermediate in the mevalonate pathway, mevalonate-5-phosphate (MVP), promotes interaction between conformational mutant p53 and DNAJA1, which inhibits ubiquitination and proteasomal degradation of mutant p53, leading to mutant p53 protein stabilisation. These stabilised proteins bind to SREBP2, increasing the expression of mevalonate pathway enzymes and subsequently increasing MVP levels—forming the positive feedback loop [84,86]. There is also an indication that RhoA geranylgeranylation, downstream of the mevalonate pathway, also stabilises mutant p53, as well as by RhoA- and actin-dependent transduction of mechanical inputs [87].

Statins, as inhibitors of the mevalonate pathway, also block or reverse many of the pro-carcinogenic effects of mutant p53. It has been shown that statins reduce mutant p53 levels in vitro by reversing the stabilisation of mutant p53 via action on the mevalonate pathway [84,87]. Similarly, statin treatment reduces metabolic plasticity by reducing increases in lipid metabolism and cholesterol synthesis.

Intriguingly there are also indications that many of the ‘off-target’ effects of statins may reduce elements of the pre-cancer niche and thereby reduce carcinogenesis in LFS. Data from studies in cardiovascular disease show that statin treatment may reduce the risk of cardiovascular disease by targeting telomere shortening, which is a risk factor [88,89]. Telomere shortening, exacerbated by mitochondrial dysfunction and oxidative stress, causes DNA damage, cellular senescence and further inflammation in cardiomyocytes, ultimately leading to cardiovascular events. This is essentially the same ‘accelerated aging’ mechanism as we posit causing carcinogenesis in LFS. A clinical trial assessed the impact of statins on the relationship between telomere length, statins and cardiovascular events in a primary prevention trial of middle-aged men at high risk of coronary heart disease events [90]. The study showed that shorter telomere length was associated with increased risk of clinical events and that treatment with pravastatin attenuated this risk. Notably, these results were independent of changes in cholesterol levels, triglycerides and other markers associated with the lipid-lowering effects of statins.

Tissue-specific factors are also an important aspect of the pre-cancer niche, and here, too, the mutant p53/mevalonate pathway feedback loop is also implicated. Genome-wide analysis and 3D culture models were used to show that mevalonate pathway intermediates and mutant p53 were associated with changes to cell morphology in line with breast cancer, and that statin treatment was able to revert these phenotypic changes [91]. There is also evidence that statins inhibit cancer neo-angiogenesis [92].

While there are in vivo data showing that statin treatment (atorvastatin and rosuvastatin) reduces tumour growth in mice bearing tumours with mutant p53 [86] and appears to have positive effects in lung cancer patients with somatic p53 mutations [93], there have been no studies assessing the impact of cancer incidence in relevant murine models of LFS. Such studies are warranted given the evidence to date.

### 3.3. Aspirin

Aspirin (acetylsalicylic acid) is one of the most studied of the oncological drug re-purposing candidates, both for cancer prevention and treatment. The ReDO_Trials Database lists 39 active trials, (as of 14 December 2021) that include aspirin in a treatment arm. In terms of cancer prevention, there are numerous retrospective studies showing positive effects, particularly in colorectal cancer [94,95], but also in breast [96], liver [97] and some other cancers. There are also some data suggesting beneficial effects of aspirin post-diagnosis in colorectal and breast cancer [98,99]. The impact of aspirin on the reduction of cancer recurrence after curative treatment of breast, colorectal, gastro-oesophageal or prostate cancer is currently the subject of a large, international placebo-controlled randomised clinical trial (the ADD-Aspirin trial) [100].

The anti-cancer effects of aspirin are still being elucidated, but it is one of the few agents which has evidence to show that it addresses all of the ‘hallmarks of cancer’ [101]. Succinct reviews of the relevant mechanisms of action related to cancer prevention in different populations/cancers are provided by [102,103,104]. Despite the wealth of data on the multiple anti-cancer mechanisms of action, very little attention has been focused on the direct relationship between aspirin and p53, and still less on aspirin and LFS.

However, it has been shown that aspirin acetylates p53 in both wild-type and mutant p53 [105,106]. This post-translational modification stabilised p53, increased localisation to the nucleus, increased DNA-binding activity and induced p21^CIP1^ and Bax, both of which are important in p53′s cell cycle control functions. This work, which unfortunately has not been further explored, suggests that aspirin may partially reactivate mutant p53, restoring some of the loss of functionality or reverting gain of functions. Additional studies are required to confirm these findings and to assess, in vivo, whether these effects can reduce the risk of cancer initiation in animal models of LFS.

While the direct effects of aspirin on pathogenic p53 remain relatively unexplored, there is more support for the effects on those aspects of the host environment which may be involved in malignant transformation and progression, particularly with respect to its anti-inflammatory activity [107]. Urinary PGE-M is a stable metabolite of PGE2, a key inflammatory molecule implicated in chronic inflammation and cancer, has been shown in clinical trials to be reduced by treatment with aspirin [108,109]. PGE2 is also a factor in immune dysregulation, and therefore, reducing circulating PGE2 may also impact this aspect of the pre-cancer niche [110,111]. The effect of aspirin on immunity is an active topic of research, clinical and pre-clinical, with some data to suggest that it may help to potentiate check-point inhibition and other immunotherapies [112].

Aspirin also has multiple anti-angiogenic effects [113], with some evidence from men with prostate cancer that regular low dose aspirin use is associated with a lower angiogenic phenotype [114]. Aspirin also targets metabolic pathways via activation of AMPK and inhibition of mTOR [101,115]; notably, these are also important in the anti-cancer activities of metformin, and there is some interest in exploring the potential synergy of the two drugs in cancer treatment [116,117].

Other mechanisms which may be important for cancer prevention in LFS include tissue-specific factors that encourage cancer formation in specific niches. There have, for example, been studies which have shown that aspirin may be associated with a reduction in mammographic breast density [118]. We have previously noted the similarity between the pre-cancer niche and the wound-healing response, and there has long been a suspicion, going back over a century, that sarcomas in particular may be associated with sites of physical injury or trauma [119,120]. Here, too, aspirin is of interest in that it can control an aberrant wound healing response and thereby reduce the risk of cancer initiation [121].

Finally, although aspirin has been suggested as an anti-aging agent [122], the data showing a positive effect on telomere length are still relatively scarce [123]. However, a reduction in oxidative stress with aspirin should, in theory, lead to lower rates of telomere attrition [124].

### 3.4. Propranolol

Propranolol, a classical non-selective beta blocker, is a drug that has been re-purposed multiple times and now has broad range of medical uses that extend beyond the hypertension that it was originally developed for. It is also another oncological re-purposing candidate that addresses multiple targets and pathways that are of interest in cancer. In particular, propranolol has been shown to be anti-angiogenic; to reduce cancer cell proliferation, invasion and migration; to have positive effects on anti-tumour immunity; and also to sensitise resistant cancer cells to cytotoxic treatments [125]. It also reduces PGE2, which addresses some of the downstream effects of oxidative stress [126]. For these reasons, it has been described as a possible anti-metastatic drug, and there are data showing that it reduces the rate of metastatic spread in a range of murine models; there is also some support from retrospective studies in breast cancer showing a reduction in rates of metastatic disease in women taking propranolol [127].

In addition to these general anti-cancer properties there are also several others which are of interest in the context of LFS. Psychological stress and anxiety are an inevitable consequence of a diagnosis of LFS, particularly in families with a history of cancer incidence and mortality [128]. There are also periods of intense anxiety associated with waiting for results of routine or diagnostic scans—a phenomenon termed ‘scanxiety’ by cancer and LFS patients alike [129]. It is known that psychological stress has physical manifestations that have been shown, in animal models, to be associated with cancer [130,131]. Data from numerous studies in humans have also found a negative influence of psychological stress on cancer incidence and survival [132]. These deleterious effects of stress are mediated by the neuroendocrine system, in particular via the hypothalamic–pituitary–adrenal and sympathetic nervous systems [133,134,135]. It should be emphasised that there are no direct data showing an association between levels of stress and cancer in LFS patients, but data from the general population indicate distinct physical effects of stress on immune function and inflammation that are relevant to LFS and carcinogenesis [136,137,138]. There are also data showing that psychological stress may increase oxidative stress [139]. It may be, therefore, that the psychological stressors associated with LFS may directly influence pre-cancer niche formation and/or the chronic inflammation that hastens carcinogenesis.

Blockade of beta-adrenergic signalling with propranolol has been shown to reduce many of these adverse effects of stress [138,140,141,142]. In addition to reverting the immune dysfunction and oxidative stress associated with psychosocial stressors, it would also have anxiolytic effects that may have positive outcomes on mood and coping [143].

Propranolol also has some effects on cellular metabolism which may be relevant in cancer. Data from mouse models show that it can act as a late block in the autophagic cascade via the beta2 adrenergic receptor [144,145]. Inhibition of autophagy interrupts the metabolic shuttle between cellular compartments associated with the reverse Warburg/Warburg phenotypes, thereby reducing conditions conducive to cancer. Furthermore, in a breast cancer model, propranolol reduced glucose metabolism, as evidenced by reduced (18)F-FDG PET imaging of 4T1 breast tumours [146]. Propranolol also impacts lipogenesis via inhibition of lipin-1, which is regulated by p53, again addressing an important aspect of the metabolic plasticity associated with cancer initiation and progression [147,148,149].

A final factor that makes propranolol an intriguing drug candidate is that it may have indirect effects on telomere attrition and accelerated aging. Stress has a negative impact on telomere maintenance, with evidence from a range of sources indicating that increased psychosocial stress is associated with telomere attrition, a process exacerbated by a positive feedback loop between psychological stress and reactive oxygen species, inflammation and mitochondrial dysfunction [150,151]. Propranolol has been shown to attenuate these biological downstream effects of psychosocial stress in healthy young adults [152].

While there is a need to test propranolol in animal models of LFS, there are some encouraging data that show it can reduce rates of chemically induced (4-nitroquinoline-1-oxide) carcinogenesis in murine models of oral cancer, (a model which re-capitulates some aspects of inflammatory-driven cancer via a specific tissue niche) [153].

### 3.5. Other Candidate Drugs

The four candidates listed above are interesting because they target multiple pathways and factors relevant to the pre-cancer niche and/or are supported by LFS-specific data. However, there are many other candidate drugs which are of interest but which do not address so many of these targets.

Sirolimus, a drug used to prevent transplant organ rejection, is an mTOR inhibitor also of interest as an anti-cancer drug, particularly in the treatment of perivascular epithelioid cell tumour (PEComa) [154]. It was also a drug that extended survival and reduced cancer incidence in a heterozygous p53 ± mouse model [155]. Unfortunately, there are no follow-up data in a more representative animal model of LFS to allow us to assess whether this is a candidate deserving of further investigation.

Arsenic trioxide (ATO) is the standard of care treatment for acute promyelocytic leukaemia (APL) in combination with all-trans retinoic acid [156]. Recent work by Min Lu and colleagues reports in vitro and in vivo data showing that ATO treatment is able to restore wild-type activity to a range of common p53 mutants, including many that are common in people with LFS [157]. Treatment with ATO rescued mutant p53 and restored tumour suppressive activity, leading to increased survival in mice bearing established tumours with mutant p53. These animal experiments were performed using ATO doses that delivered plasma concentrations similar to those of APL patients and are therefore considered achievable in humans. This encouraging work did not specifically use murine models of LFS, and it is hoped that such work can take place in the future. However, ATO has toxicity and is not designed for chronic use. It may be that chronic low-dose metronomic dosing can reduce toxicity and still achieve restoration of p53 function, but that has yet to be assessed. It may also be that other arsenic compounds, such as the oral drug Realgar-Indigo Naturalis Formula, which is approved for APL treatment in China [158], may also have these effects on p53 with a lower level of toxicity and a formulation that is more appropriate for long-term use. Alternatively, periodic courses of ATO over a lifetime may also serve to reduce cancer incidence in LFS. These are all speculative and dependent on additional pre-clinical research. In the meantime, ATO should be investigated clinically when treating people with LFS with cancer, particularly when those cancers are refractory or recurrent.

The bisphosphonates—including zoledronate, ibandronate and alendronate—are primarily used to treat osteoporosis and the skeletal complications of bone metastases in cancer. There is convincing evidence that they reduce the risk of breast cancer recurrence, and the most recent ASCO-OH guidelines recommend adjuvant clodronate, ibandronate and zoledronic acid for this indication [159]. The nitrogenous bisphosphonates, which includes zoledronate, ibandronate and alendronate, are also candidates for further pre-clinical investigation in mouse models of LFS. Nitrogen-containing bisphosphonates target the mevalonate pathway similarly to the statin class of drugs [160,161], and in some studies both statins and bisphosphonates have shown the same types of activity with respect to effects on mutant p53 [87]. There is some evidence that bisphosphonates also affect metabolic pathways, including lipogenesis [162], including retrospective data in osteoporosis and Paget’s disease patients showing that zoledronate reduced circulating blood glucose and atherogenic lipids [163]. Many of the mechanisms by which these drugs reduce the risk of metastatic disease and recurrence may also be relevant to the process of malignant transformation, but the data for this in LFS models are lacking. Of the different drugs in this class, it is the oral bisphosphonates, such as ibandronate and alendronate, which may be of most interest in a cancer-prevention context, and therefore, any pre-clinical work specifically for LFS should focus on them.

There are, of course, numerous other drug candidates which have one or more mechanisms of action which make them interesting for cancer prevention in LFS. Re-purposing examples specifically targeting mutant p53 include verteporfin, which activated p73 in mutant p53 pancreatic cancer cells [164,165], mebendazole [166,167] and valproic acid [168,169]. There are also many drugs which target chronic inflammation and metabolic dysfunction and which are used at chronic dosing and may therefore also be interesting, for example, drugs in the NSAID class such as diclofenac [170].

Of course, re-purposing is not the only option, and there are a number of small molecule drugs being developed to restore wild-type activity in mutant p53 [171]. A recent review of such strategies in head and neck squamous cell carcinoma, a malignancy with high rates of somatic TP53 mutations, included the agents PRIMA-1, APR-246, RITA, COTI-2 and CP-31398 [172]. Another interesting avenue of exploration is zinc metallochaperones (ZMCs)—small molecule agents that reactivate zinc-deficient p53 mutants [173].

An alternative approach is the development of drugs seeking to capitalise on p53 mutants through a synthetic lethality strategy that targets downstream kinases such as WEE1 and CHK1 to cause mitotic catastrophe in DNA-damaged cells [174]. The first trials of WEE1 inhibitors have already shown encouraging signals of efficacy in patients bearing p53-mutant cancers, although not without toxicity [175,176,177]. These agents are being developed in the context of tumours carrying somatic mutations; the effect on the pre-cancer niche in animal models has not been explored, although it is an intriguing prospect. However, the toxicity of these new agents currently means that chronic use as cancer prevention agents is not feasible. A more fruitful development for them may be as anti-cancer therapeutics which are directed at LFS patients who go on to develop cancers. In this scenario we can seek molecularly targeted cancer-prevention agents for people with LFS, as proposed in this paper, which can be complemented by molecularly targeted anti-cancer agents should cancers arise at a later point.

## 4. Discussion

In this paper, we have reviewed the evidence supporting the re-purposing of a number of drug candidates as cancer prevention treatments for people with LFS. Factors relevant to the selection of the candidates depend in large part on mechanisms of action that inhibit aspects of the priming for cancer that is a consequence of germ-line mutation in TP53. The pre-cancer or premalignant niche is a site in which high rates of oxidative stress, the circulation of pro-angiogenic factors, dysregulated immunity, aberrant cellular metabolism and/or tissue-specific events (e.g., injury, elevated levels of stress hormones, aromatase, etc.), come together to create a local environment conducive to malignant transformation and cancer progression [18,33]. The candidate drugs we have reviewed all interfere with multiple steps in this process—reducing oxidative stress, inhibiting metabolic plasticity, reverting immune dysfunction, etc., as shown in Figure 4.

To date, very few interventions have been shown to reduce cancer incidence in animal models of LFS. In addition to metformin and sirolimus, which have been discussed above, the other intervention which has been shown to reduce cancer incidence in a murine LFS model is a calorie-restricted diet [178]. In this study, adult heterozygous p53 mice were randomised to receive a standard diet, a calorie-restricted (CR) diet (60% of standard) or a one day/week fast (F). Mice in the CR and F groups had improved overall mean survival compared to the control group, 388 and 357 days versus 313 days. Both the 24% increase in the CR group compared with control mice and the 14% increase in fasted mice compared to controls were statistically significant (*p* = 0.001 and *p* = 0.039, respectively). Biomarkers associated with this improvement in survival were assessed in a group of CR mice, and it was shown that plasma IGF-1 and leptin were significantly reduced compared to controls, both *p* < 0.05.

The link between CR and life extension is well-known and appears to be conserved across a number of species [179]. Key mechanisms in this process are also relevant to cancer incidence in the general population and not just in people with LFS [180,181], specifically, improvements in mitochondrial efficiency, reductions in chronic inflammation and oxidative stress, activation of AMPK and down-regulation of mTOR [180,182]. Calorie restriction of the type that was used in the animal model of LFS, that is, a 60% decrease in calorific input, are hard to achieve consistently in humans, and therefore there is intense activity in seeking calorie-restriction mimetics—agents which can reproduce the main biological effects of CR [183]. It is notable that metformin, aspirin, sirolimus, statins and propranolol are considered as candidate CR mimetics or anti-aging agents [183,184].

Inevitably, these findings raise questions about the fundamental origins of cancer—and it suggests that mitochondrial dysfunction plays a central role in carcinogenesis. While it is too early to suggest, as some have, that cancer in general is a mitochondrial/metabolic disease [185,186,187], the evidence that it is an important driver of carcinogenesis in LFS is strong. It may also be that telomere attrition, which is common in LFS, as we have outlined previously, may also be a consequence of that mitochondrial dysfunction [188].

These findings also raise questions about non-medical interventions, for example, are there diets or lifestyle alterations that can re-capitulate the reductions in chronic inflammation, activation of AMPK and other effects described above? We know for our interactions with the LFS community that there is a huge interest in this topic, and many people are attempting to make healthy choices, but there are few hard data to support any particular lifestyle or dietary choice. One hope is that the MILI trial will provide some evidence that may be used to help answer such questions in the future.

## 5. Conclusions

The candidate drugs we have reviewed in this paper each address multiple pathways that may inhibit the creation of pre-cancer niches, reduce the incidence of malignant cell transformation and/or reduce the chances of a transformed cell giving rise to progressive disease. However, it may well be that there is no single drug that can block every step of the complex cascade that results in a viable tumour. We may speculate, therefore, that a combination of these candidate drugs may be more effective than any single agent in stopping germline TP53 mutations from acting as the ‘guardian of the cancer cell’ [189]. A ‘polypill’ that combines some of these drugs into a single capsule is some way off, but such a rational combination is certainly deserving of further consideration.

In the meantime, the LFS community welcomes the MILI trial, which is a first step in the process of discovering how to stop the priming of cancer that is a defining feature of LFS.

## Figures and Tables

**Figure 1 cancers-14-01621-f001:**
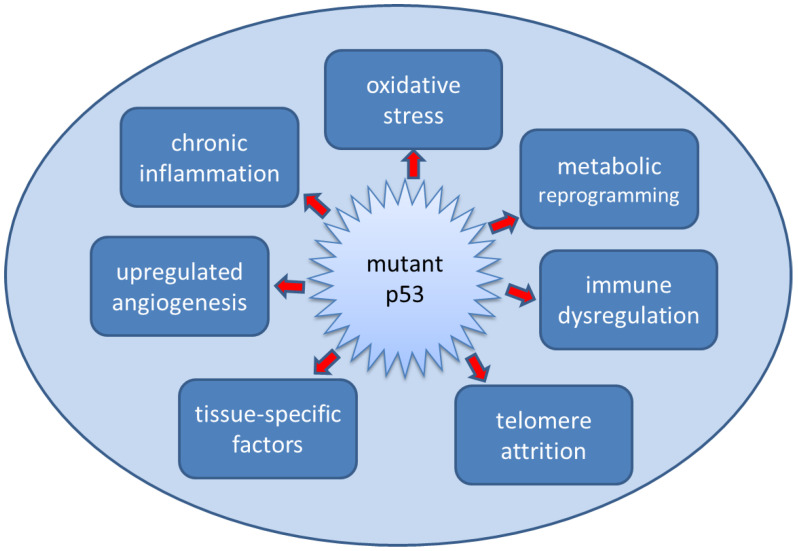
Mutated p53 function leads to the creation of a pre-cancerous niche. Mutant p53 drives the creation of specific biological niches in which chronic inflammatory responses, including elevated basal oxidative stress, metabolic reprogramming, the release of pro-angiogenic factors and immune dysregulation combine to create the conditions conducive to malignant transformation. Increased oxidative stress and/or telomere attrition and tissue-specific factors, for example increased aromatase expression in the breast, contribute to additional genetic events leading to cancer initiation that arises in these cancer-supporting niches.

**Figure 2 cancers-14-01621-f002:**
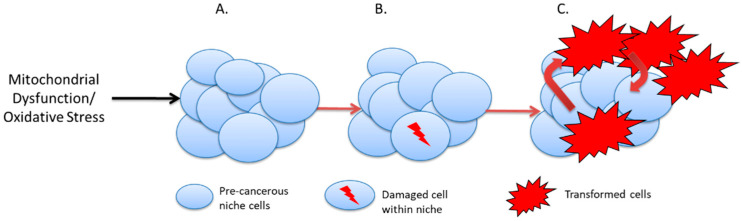
Cancer initiation in the pre-cancer niche. (**A**). Cells within the precancerous niche undergo telomere attrition or suffer genetic damage due to elevated oxidative stress. (**B**). Telomere crisis or further DNA damage may lead to loss of heterozygosity and malignant transformation. (**C**). Malignant cells in contact with chronically inflamed pre-cancerous niches proliferate and initiate tumour growth.

**Figure 3 cancers-14-01621-f003:**
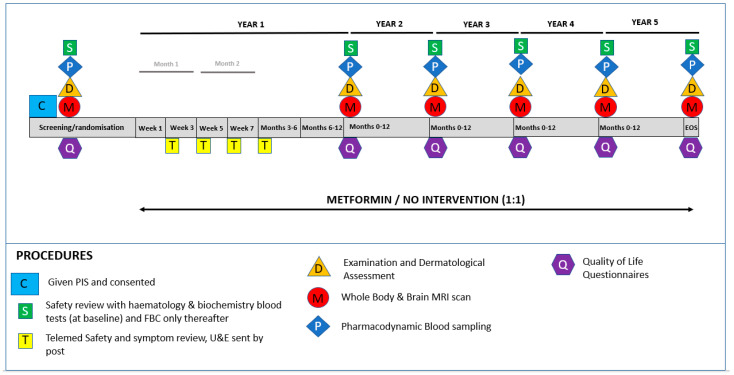
MILI trial schema.

**Figure 4 cancers-14-01621-f004:**
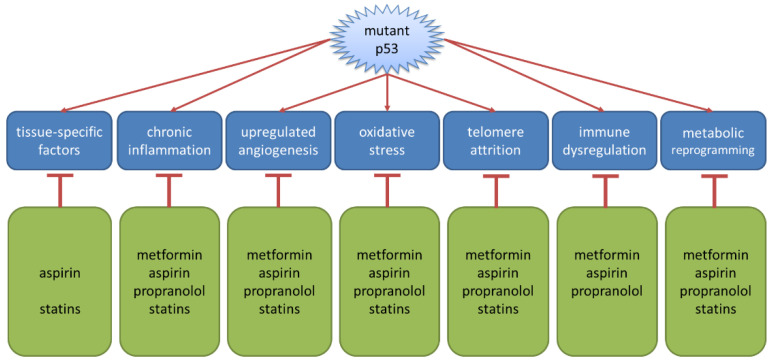
Candidate drugs inhibit key facets of the pre-cancer niche to reduce the risk of malignant transformation.

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
