# Peer review of "Inhibiting the Priming for Cancer in Li-Fraumeni Syndrome"

_cancers, 2022, doi:10.3390/cancers14071621_

Round 1

Reviewer 1 Report

The present manuscript by Pantziarka et al, faces a difficult task in that he first attempts to explain the pre-cancerous niche of the Li-Fraumeni syndrome. He then reviews possible existing compounds in a process of drug-repurposing focused on Li-Fraumeni syndrome. Both of them are themes that the authors have described extensively in previous publications.

I have read with great interest this manuscript and I found it overall well written. I have a few minor observations or comments to make:

1 - I found close similarities of the present manuscript with previous publications from the same group. In particular two figure and their legends of the present manuscript are identical to those in a similar work published  in 2015 [Ecancermedicalscience 2015;9,541]. It could be helpful to modify them a bit or refer to the original publication.

2 - This manuscript is intended to be published in a special issue on the role of TP53 in cancer which will probably include several graphical representations of the TP53 pathway. However, I think that a picture drawing the different points of attack of the cited repurposed drugs in the TP53 pathway could be useful and interesting.

3 -  The title of this manuscript refers to inhibiting the priming in Li-Fraumeni syndrome and reviews several known existing drugs. However, it cannot be forgot that new emerging drugs (such as Wee1 inhibitors) already exist and they should be cited also in the perspective of Li-Fraumeni pre-cancerous niche. Could old and new drugs could interact with each other and how? 

Author Response

Thank you for the useful feedback and observations on this paper. Below are listed point by point responses to these.

The present manuscript by Pantziarka et al, faces a difficult task in that he first attempts to explain the pre-cancerous niche of the Li-Fraumeni syndrome. He then reviews possible existing compounds in a process of drug-repurposing focused on Li-Fraumeni syndrome. Both of them are themes that the authors have described extensively in previous publications.

I have read with great interest this manuscript and I found it overall well written. I have a few minor observations or comments to make:

1 - I found close similarities of the present manuscript with previous publications from the same group. In particular two figure and their legends of the present manuscript are identical to those in a similar work published  in 2015 [Ecancermedicalscience 2015;9,541]. It could be helpful to modify them a bit or refer to the original publication.

Figures 1 and 2 have been reworked and further refined to emphasise points made in this paper. The changes are more than cosmetic, and it is hoped that they more fully represent what is new compared to previous exposition of some of the ideas also presented in this publication.

2 - This manuscript is intended to be published in a special issue on the role of TP53 in cancer which will probably include several graphical representations of the TP53 pathway. However, I think that a picture drawing the different points of attack of the cited repurposed drugs in the TP53 pathway could be useful and interesting.

A new figure (fig 4) has been introduced to summarise the effect of the different candidate drugs on the downstream effects of mutant p53. This figure shows both commonalities among the candidate drugs and also differences where they exist.

3 - The title of this manuscript refers to inhibiting the priming in Li-Fraumeni syndrome and reviews several known existing drugs. However, it cannot be forgot that new emerging drugs (such as Wee1 inhibitors) already exist and they should be cited also in the perspective of Li-Fraumeni pre-cancerous niche. Could old and new drugs could interact with each other and how?

A new paragraph has been introduced at the end of section 3 to discuss WEE1 and CHK1 inhibitors and to outline how these agents might fit in with cancer prevention and treatment of people with LFS. This is an important topic and the authors are grateful for the opportunity to include them in this review.

Reviewer 2 Report

This is a relevant updated review on LFS, that emphasized the potential carcinogenesis of TP53 mutational status, while druggability of TP53 remains future challenges. The TP53 germ-line mutation carriers has a substantial clinical importance for appropriate method of cancer surveillance and counselling for this patients. Another interesting point are somatic mutation of TP53 as in rare germinal tumors. To date, this impact of prognosis of testicular rare cancer and chemotherapy sensitivity, have to be better investigated.

Author Response

Thank you for these useful comments. 

A new paragraph has been introduced into the introduction to discuss the point that what we can learn from LFS might also generalise to spontaneous cancers in the wider population, particularly for cancers in which somatic TP53 mutations are common, including resistant/recurrent testicular cancer.